# A Y chromosome-linked genome editor for efficient population suppression in the malaria vector *Anopheles gambiae*

Ignacio Tolosana [1], Katie Willis [1], Matthew Gribble[1], Lee Phillimore[1], Austin Burt [1], Tony Nolan [2] ✉, Andrea Crisanti [1] & Federica Bernardini [1] ✉

Genetic control – the deliberate introduction of genetic traits to control a pest or vector population – offers a powerful tool to augment conventional mosquito control tools that have been successful in reducing malaria burden but that are compromised by a range of operational challenges. Self-sustaining genetic control strategies have shown great potential in laboratory settings, but hesitancy due to their invasive and persistent nature may delay their implementation. Here, instead, we describe a self-limiting strategy, designed to have geographically and temporally restricted effect, based on a Y chromosome-linked genome editor (YLE). The YLE comprises a CRISPR-Cas9 construct that is always inherited by males yet generates an autosomal dominant mutation that is transmitted to over 90% of the offspring and results in female-specific sterility. To our knowledge, our system represents a pioneering approach in the engineering of the Y chromosome to generate a genetic control strain for mosquitoes. Mathematical modelling shows that this YLE technology is up to seven times more efficient for population suppression than optimal versions of other self-limiting strategies, such as the widely used Sterile Insect Technique or the Release of Insects carrying a Dominant Lethal gene.

Vector control has proven to be one of the most efficient ways to reduce malaria incidence through history[1–3]. The use of insecticide-treated bed nets (ITNs) and indoor residual spraying (IRS) have driven the reduction of malaria cases since the year 2000[1,2]. However, different factors such as challenges in the use and distribution of bed nets, the rise of resistance towards insecticides and antimalarial drugs, and the behavioural plasticity of the mosquito vectors have contributed to the decrease in the progress against malaria since 2015, highlighting the need for new tools and strategies to control the disease[2].

Genetic control of vector populations arises as a very promising field in this framework, as it offers a range of technologies with diverse efficacy and control prospects but with the common characteristic of being species-specific and environmentally non-polluting[4,5]. These strategies can be divided into those that aim at modifying the mosquitoes' genome to prevent the development of the malaria parasite[6,7] and those that aim at suppressing mosquito populations[8–11]. Genetic approaches for vector control also differ in their intended post-release dynamics: some technologies are designed to increase in frequency from a small release and spread geographically (self-sustaining strategies), while others pose a temporal and spatial restriction on the persistence of the modification (self-limiting) and require large and continuous releases to maintain their effect on the target population[4,12].

Traditionally, the Sterile Insect Technique (SIT) has relied on the release of radiation-sterilised males, and it has been widely applied for insect pest control programmes[13–15]. More recently, transgene-based SITs and other alternative self-limiting technologies based on the

[1]Department of Life Sciences, Imperial College London, London, UK. [2]Department of Vector Biology, Liverpool School of Tropical Medicine, Liverpool, UK.
✉e-mail: tony.nolan@lstmed.ac.uk; f.bernardini11@imperial.ac.uk

presence of lethality-inducing transgenes such as the Release of Insects carrying a Dominant Lethal gene (RIDL), and on synthetic sex ratio distortion systems capable of inducing a strong male bias in the progeny have been developed and tested in the lab or in the field for different vector species[10,16–21]. While these strategies have been useful in certain contexts, they share the crucial requirement of large-scale releases. To effectively reduce malaria incidence, which is largely a rural disease, the high rate of male mosquito releases, together with the associated cost of mass production, make the implementation of such technologies challenging, especially in extensive areas of sub-Saharan Africa where the malaria burden is by far the highest.

To overcome this, Burt and Deredec suggested the design of a self-limiting genetic tool, termed Y chromosome-linked genome editor (YLE)[22], that would, in theory, have a more prolonged suppression effect in the target insect population. This technology is based on a genetic construct located on the Y chromosome that creates mutations that induce sterility or lethality when inherited by the female offspring but have no effect in the originator male or the male offspring (Fig. 1a). The genetic element encoding the editor would not increase its frequency in the population, but assuming no fitness costs, it is expected to remain at the released frequency because it follows Mendelian inheritance and is not selected against: the YLE construct only causes harm in the female offspring, where the Y chromosome is not present, and there is no alternative allele or chromosome in the YLE-bearing males to compete with. This lack of selection against the YLE has a pivotal consequence: if releases of males carrying the transgene are halted, the population would not immediately recover (as occurs with many other self-limiting strategies such as SIT, RIDL, or a female-specific RIDL), but the level of initial suppression would remain indefinitely[22] (Fig. 1b).

These features make YLE technologies of great interest; however, the development of genetic strategies to control *Anopheles* mosquitoes that rely on the Y chromosome have proven to be challenging. This is mainly due to the highly heterochromatic nature of this chromosome and to the abundance of repetitive sequences that have prevented a complete chromosome assembly, hampering precise genome manipulation and site-specific transgene integration[23–25]. Furthermore, transgenes inserted on the Y chromosome of *Anopheles*

species exhibit silencing due to heterochromatin and variable levels of expression due to position-effect variegation (PEV), as occurs in *Drosophila*[23,26,27]. Besides these difficulties, meiotic sex chromosome inactivation (MSCI) is a major barrier to the expression of Y-linked genes in the germline, which is a necessary pre-requisite for a YLE[28–30]. Nonetheless, germline expression of Y chromosome-linked transgenes has been previously achieved in *An. gambiae* using transcriptional control sequences that are activated prior to meiosis[23,31].

Here, we use this knowledge to create a Y chromosome-linked Cas9-based system that is active in the male germline of *An. gambiae* mosquitoes. Furthermore, we discover and isolate an 11-base pair deletion in the *doublesex* gene that produces dominant female-specific sterility. Finally, we demonstrate that the Y-linked Cas9 can bias the inheritance of this deletion at a high rate, by the process of nuclease-induced homing[5]. To our knowledge, this represents proof of principle for the first genetic control strategy linked to the Y chromosome for the control of *An. gambiae* populations, and mathematical modelling data show that this technology is considerably more efficient than other self-limiting approaches such as SIT and various transgenic versions thereof. We provide experimental confirmation of these modelling predictions by showing suppression in caged populations with the release of YLE males.

Y-linked editors such as the one we develop may achieve efficient local suppression with a low risk of invading neighbouring populations, thereby overcoming the potential regulatory hurdles that more invasive strategies might face. Thus, this technology can be used both as a confined method for vector control and as a middle step in a phased malaria control programme aiming at using self-sustaining strategies.

## Results
### Y chromosome-linked Cas9 activity leads to high levels of homing

In the attempt to build a YLE technology, we first investigated Cas9 activity from the Y chromosome by designing a genetic construct (YLE$^{dsx}$) containing a gene coding for the endonuclease Cas9 under the control of the germline promoter *vas2*[31], and a gRNA, ubiquitously expressed, targeting the female-specific exon of *doublesex* (*dsxF*)

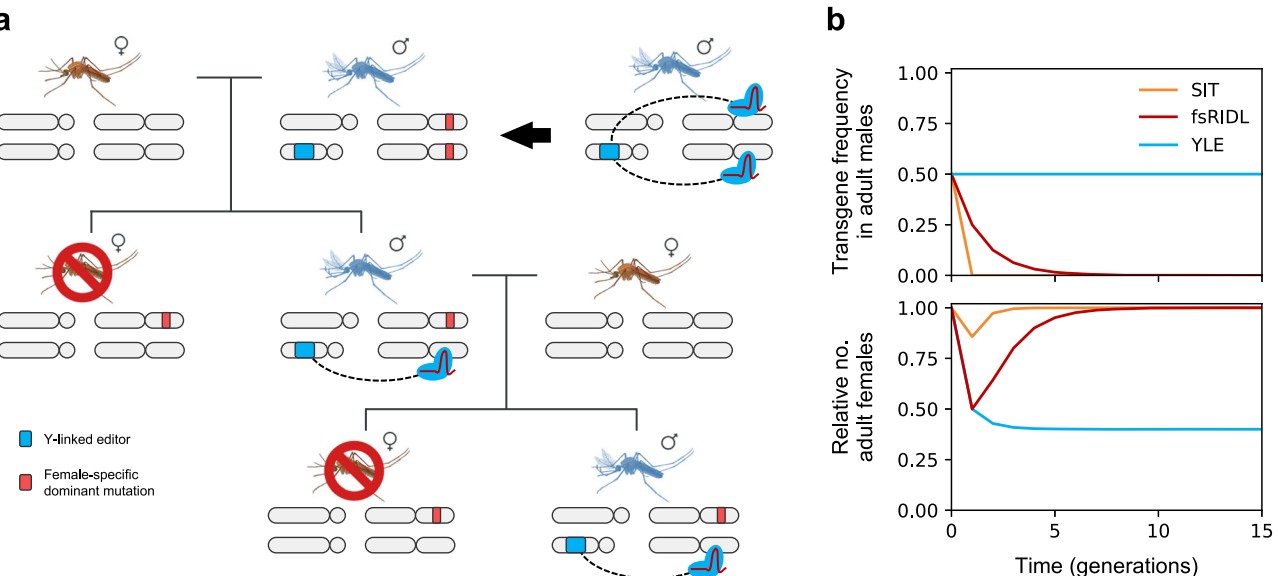

**Fig. 1 | Dynamics of the Y chromosome-linked genome editor strategy.**
**a** Schematic representation of a Y chromosome-linked genome editor construct (in blue) that generates dominant mutations in genes required for female survival/fertility (in red) in the male germline through nuclease activity. These mutations are detrimental when inherited by females (causing lethality or sterility) but do not affect males. Because the genetic construct is located on the Y chromosome, which is always

inherited by males, all the males in the offspring inherit it, allowing this process to be maintained through generations. **b** Modelling of the transgene frequency in adult males (top) and relative number of adult females (bottom) in a target mosquito population upon a single male release equivalent to 50% of the initial male population for Sterile Insect Technique (SIT, orange), Release of Insects carrying a female-specific Dominant Lethal gene (fs-RIDL, red) and Y-linked editors (YLE, blue).

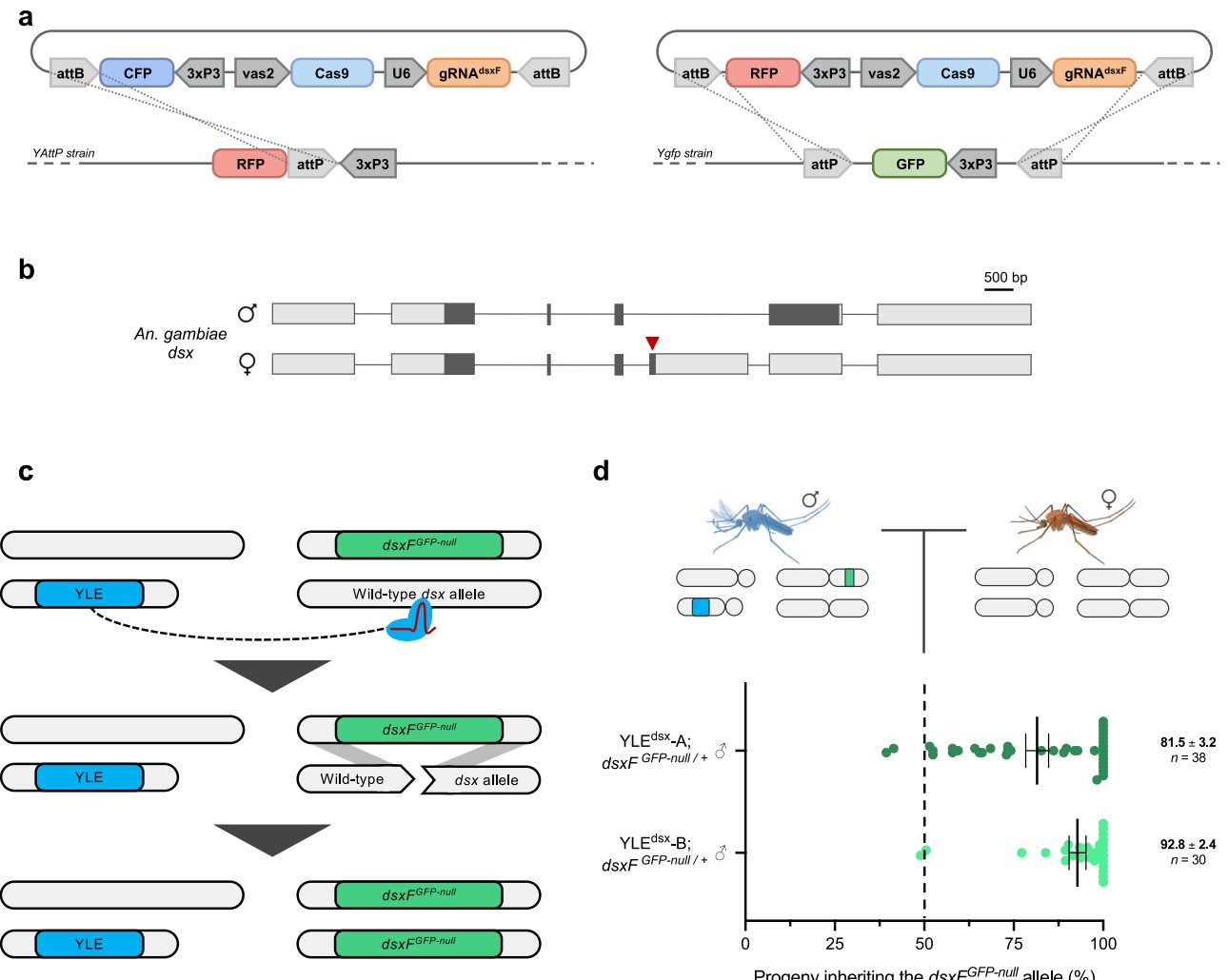

**Fig. 2 | Generation of a Y chromosome-linked genome editor and evaluation of Y-linked Cas9 activity. a** Genetic constructs used to generate a YLE based on Cas9 expression under the germline *vas2* promoter, targeting the female isoform of the *doublesex* gene. The constructs contained a gene coding for either CFP or RFP under the *3xP3* promoter and were inserted in the Y-linked loci on the YAttP and the Ygfp strains using integrase-mediated recombination. **b** Schematic representation of the *An. gambiae doublesex* gene, showing the male isoform (top) and the female isoform (bottom). Boxes represent exons, and lines represent introns. CDS and UTRs are shown in dark and light colours, respectively. Introns are not drawn to scale. The red arrow indicates the site targeted by the YLE. **c** Illustration of homology-directed repair (homing) with a modified *dsx* allele ($dsxF^{GFP-null}$) used to evaluate the Cas9 activity of the $YLE^{dsx}$ strains. **d** Percentage of progeny from $YLE^{dsx}$-A males (dark green) and $YLE^{dsx}$-B males (light green) that inherited the $dsxF^{GFP-null}$ allele when crossed to wild-type individuals ($n \geq 30$). Each dot corresponds to the progeny of a different mating couple. The dashed line indicates the expected Mendelian inheritance. Numbers on the right indicate the mean, the s.e.m., and the size of each sample (*n*). Source data are provided as a Source Data file.

(Fig. 2a, b). The $YLE^{dsx}$ construct was integrated in the genome of two independently originated *An. gambiae* strains that each harboured 'docking' sites for site-specific recombination at different locations on the Y chromosome (YAttP[23] and Ygfp[32] to generate $YLE^{dsx}$-A and $YLE^{dsx}$-B, respectively).

Our readout for Y-linked Cas9 activity was homing[5]: Cas9-induced homology-directed repair to a specific allele at the *dsx* locus, which gives a scoreable phenotype. By using a strain that contained a *GFP* insertion that creates a null allele of *dsxF* ($dsxF^{GFP-null}$)[8], we were able to detect the number of individuals that inherited the homed $dsxF^{GFP-null}$ allele in the progeny of $YLE^{dsx}$ males (Fig. 2c). We observed rates of inheritance of the $dsxF^{GFP-null}$ allele much higher than those expected from Mendelian inheritance, in both $YLE^{dsx}$-A (81.5% inheritance) and $YLE^{dsx}$-B (92.8% inheritance) backgrounds (Fig. 2d), indicative of high levels of Y chromosome-linked nuclease activity in the germline. While the results obtained for the two strains are not significantly different ($P = 0.0694$; Mann-Whitney test), higher variability was observed in Cas9 activity for strain $YLE^{dsx}$-A, suggesting that this locus might be

more exposed to position-effect variegation and thus a higher variability in transgene expression. Consistent with this hypothesis, in the offspring of $YLE^{dsx}$ males with a lower frequency of $dsxF^{GFP-null}$, sequencing of the *dsx* target site showed very low level of mutations resulting from non-homologous end joining (NHEJ) repair mechanisms, ruling out the possibility that reduced homing could have been the result of the emergence of alleles resistant to Cas9 cleavage[9,33] (Supp. Table 1).

**An 11-base pair deletion in the coding sequence of *doublesex* causes female dominant sterility**

When carrying out routine mosquito husbandry activities in the offspring of $YLE^{dsx}$ males, we observed a high frequency of genetic females showing an intersex phenotype, resembling that found in females homozygous for the null mutation in the female-specific isoform of *dsx* ($dsxF^{GFP-null}$)[8]. However, in our experimental design, Cas9-induced mutations generated in the reproductive organs of $YLE^{dsx}$ males were expected to be present in heterozygosity in the progeny.

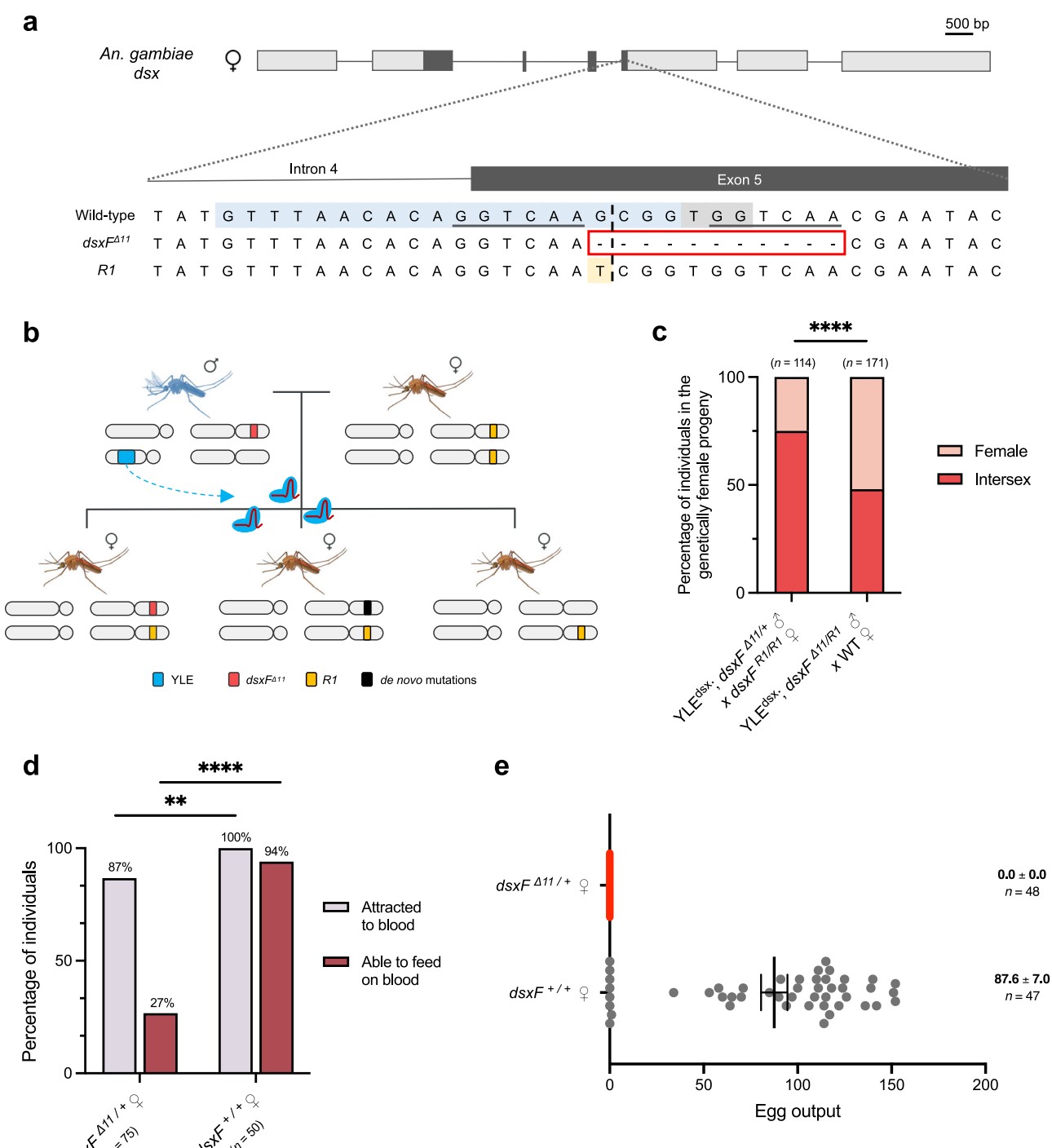

Based on the knowledge that in *An. gambiae dsx* is a haplo-sufficient gene and that the disruption of the female-specific exon is recessive[8], we considered two most likely scenarios to explain the phenotype retrieved in our experiments: 1) conversion of both *dsx* wild-type alleles into a null mutation could have occurred in females with an intersex phenotype as a result of both germline activity of Cas9 in the transgenic male progenitors and paternally deposited nuclease; or, 2) based on the knowledge that sex-specific dominant negative mutations in the *dsx* gene have been identified in *Drosophila*[34–36], a dominant negative mutation in the *dsx* gene was generated that is able to produce an intersex phenotype in females when in heterozygosity.

Consistent with the second hypothesis, amplicon sequencing of the *dsx* target locus in pools of genotypic females showing the intersex phenotype and Sanger sequencing in single individuals (over 400

individuals in total) revealed the presence in heterozygosity of an 11-base pair deletion within the coding sequence of the *dsx* female-specific exon (referred to as *dsxF^{Δ11}* hereafter) (Fig. 3a). In all the pools of intersex individuals, ~50% of the reads corresponded to the *dsxF^{Δ11}* allele and ~50% to the wild-type *dsx* allele, a result that would not be expected if stochastically created de novo mutations from deposited nuclease were responsible for the intersex phenotype. By contrast, among the phenotypic females (over 200 individuals in total), the vast majority of sequences (~99%) were the wild-type *dsx* allele, and the *dsxF^{Δ11}* mutation was present at very low frequency (~1%) (Supp. Figure1). The low frequency at which this deletion is found in this pool of females may indicate either a non-complete penetrance of the *dsxF^{Δ11}* mutation or a misidentification at the pupal stage of these individuals when the phenotype was classified.

**Fig. 3 | The dominant nature of the *dsxF^{Δ11}* allele and its impact on the biting ability and fertility of female individuals. a** Intron 4-exon 5 boundary in the *dsx* gene. The top sequence represents the wild-type allele. The gRNA target is highlighted in light blue, and the PAM is shown in light grey. A dashed line represents the expected cut site. Six-base pairs-long sequences flanking the cut site that could trigger microhomology-directed repair to generate the *dsxF^{Δ11}* allele are underlined in black. The sequence in the middle shows the *dsxF^{Δ11}* allele (highlighted with the red box). The sequence at the bottom represents the *R1* allele (SNP is highlighted in light yellow). Introns are not drawn to scale. **b** Schematic representation of the experimental cross performed with YLE^{dsx} males carrying the *dsxF^{Δ11}* allele (red) and females homozygous for the functional and resistant *R1* allele (yellow). If the Cas9-gRNA complex is paternally deposited in the embryo, the *R1* allele would prevent cleavage, and the persistence of an intersex phenotype would support the idea of the dominant nature of the *dsxF^{Δ11}* allele. New mutations potentially generated in the YLE^{dsx} males' germline are shown in black. **c** Percentage of wild-type-looking females (light red) and intersex individuals (bright red) in the progeny of YLE^{dsx};

*dsxF^{Δ11/+}* males and *dsxF^{R1/R1}* females (*n* = 114), and in the progeny of YLE^{dsx} males heterozygous for the *dsxF^{Δ11}* and *R1* alleles mated with wild-type females (*n* = 171). Statistical differences were analysed by performing a two-sided Fisher's test on the number of individuals in a 2×2 contingency table (*P* < 0.0001). **d** Percentage of intersex females carrying the *dsxF^{Δ11}* mutation (left) attracted to the blood source (grey) and able to feed on blood (brown) compared to wild-type females (right). Numbers in parentheses show the size of each sample (*n*). Statistical differences were analysed by performing a two-sided Fisher's test on the number of individuals in 2×2 contingency tables for each of the outcomes observed: attraction to blood (*P* = 0.0058) and ability to feed on blood (*P* < 0.0001). **e** Females with an intersex phenotype bearing a copy of the *dsxF^{Δ11}* allele and females with a wild-type phenotype were mated with wild-type males and subsequently were given access to a blood meal. Each dot represents the progeny of each female. Numbers on the right show the mean, the s.e.m., and the size of each sample (*n*). Source data are provided as a Source Data file.

To further confirm that the intersex phenotype was solely caused by the presence of the *dsxF^{Δ11}* allele in heterozygosity, we used a recently generated strain with a single SNP within the *dsx* target sequence that is functional and resistant to cleavage (*R1* allele)[37] (Fig. 3a). YLE^{dsx} males carrying the *dsxF^{Δ11}* allele (YLE^{dsx}; *dsxF^{Δ11/+}*) were mated with females homozygous for the *R1* allele. This would result in offspring harbouring the maternally inherited *dsx R1* allele and the paternally inherited one (*dsxF^{Δ11}*, another mutation, or wild-type *dsx*) (Fig. 3b). If the intersex phenotype were ordinarily caused by the activity of deposited Cas9 and the generation of additional recessive mutations in the maternally inherited wild-type allele, the presence here of the functional *R1* allele that prevents cleavage should lead to an absence of the intersex phenotype amongst the genetically female progeny. On the contrary, we observed that 75% of this progeny displayed an intersex phenotype (Fig. 3c). Amplicon sequencing at the *dsx* locus in these intersex individuals confirmed that both the *R1* and the *dsxF^{Δ11}* alleles were found in similar frequencies, nearing 50%, while *dsxF^{Δ11}* was absent in the pool of wild-type females, supporting the notion that *dsxF^{Δ11}* is a dominant negative allele (Supp. Figure 2). As a control, mating of YLE^{dsx} males carrying both the *dsxF^{Δ11}* and the *R1* alleles with wild-type females resulted in 50% of the female progeny inheriting the *R1* allele and showing a wild-type phenotype, and 50% inheriting the *dsxF^{Δ11}* allele and presenting an intersex phenotype (Fig. 3c, Supp. Figure 2), confirming both resistance of the *R1* allele to cleavage and absence of significant paternal deposition of Cas9.

Next, we conducted further analysis to explore the phenotype of intersex females. Dissections of 15 adult individuals revealed the presence of male-specific traits, including incompletely formed claspers and organs resembling male accessory glands in conjunction with female-like antennae and apparently normal ovaries and spermathecae (Supp. Figures 3, 4). Most importantly, after allowing intersex females to mate with wild-type males and providing a blood meal, only 27% of them were able to feed on blood (compared to 94% in the control), and all of them failed to lay eggs, not only confirming full sterility for these individuals but also an important reduction on their biting ability (and therefore vectorial capacity) (Fig. 3d, e).

### Building a YLE strategy that biases the inheritance of the female-specific dominant mutation in the progeny

The existence of female-specific dominant negative mutations at the *dsxF* locus in *An. gambiae* is especially relevant for the strategy described here because it allows the possibility to use *dsx* as a target of a YLE to induce sterility in the female offspring. Because the *dsxF^{Δ11}* allele is unrecognisable by the gRNA present in the YLE^{dsx} construct, this dominant mutation could be integrated in a system that biases the inheritance of *dsxF^{Δ11}* by inducing cleavage of the wild-type allele on the homologous chromosome from a source of Cas9 on the Y chromosome, and its conversion, through homology-directed repair, to the *dsxF^{Δ11}* allele ('homing').

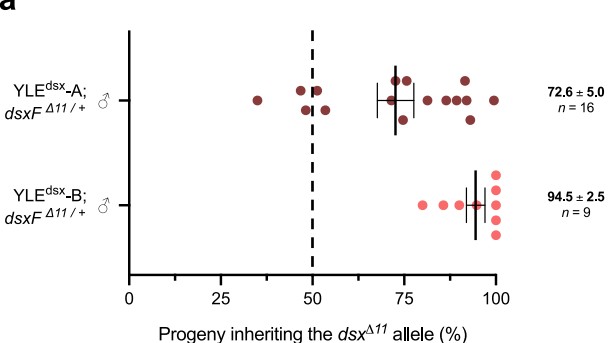

**a**

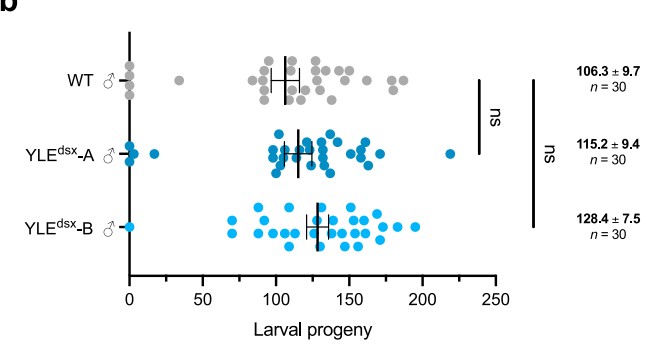

**b**

**Fig. 4 | Transmission rate of the *dsxF^{Δ11}* allele and fertility of YLE^{dsx} males. a** Percentage of progeny from YLE^{dsx}-A males (maroon) and YLE^{dsx}-B males (red) that inherited the *dsxF^{Δ11}* allele when mated with wild-type individuals. Each dot corresponds to the progeny of a different mating couple. Dashed line indicates the expected Mendelian inheritance. Numbers on the right indicate the mean and the s.e.m., and the size of each sample (*n*). **b** YLE^{dsx} males from both strains (**A** and **B**) were mated with wild-type females. A Kruskal-Wallis test (and a two-sided Dunn's multiple comparisons test with the wild-type males group as a control) revealed no significant differences in the number of larvae produced by females mated with YLE^{dsx} males and those mated with wild-type males (*P* = 0.2322 for the Kruskal-Wallis test, *P* = 0.8407 and *P* = 0.1753 for Dunn's test for comparison between the wild-type males group and strains A and B respectively). Each dot represents the progeny of each female. Numbers on the right-hand side show the mean and the s.e.m., and the size of each sample (*n*). Source data are provided as a Source Data file.

We found that the *dsxF^{Δ11}* allele was inherited by an average of 72.7% and 94.5% in the offspring of YLE^{dsx}-A and YLE^{dsx}-B males, respectively (Fig. 4a). Consistently, females inheriting this mutation showed the intersex phenotype. New mutations at the *dsx* locus were found at very low frequencies (at an average of 1% and 3% of the individuals in the progenies of YLE^{dsx}-A and YLE^{dsx}-B males

respectively), similar to what we observed in the experiments conducted using the *dsxF*[GFP-null] allele (Supp. Tables 2-3). In one of these sets of progenies, in which all the sibling females showed an intersex phenotype, we identified a one-base pair deletion within the coding sequence of exon 5 present in heterozygosity in 28.4% of the individuals, which may represent an additional dominant mutation (Supp. Table 3, Supp. Figure 5).

### The YLE[dsx] transgene does not pose a fertility cost on male mosquitoes

The YLE[dsx] strains described in this study produce progeny consisting of sterile females and transgene-carrying males. To be most useful as a genetic control tool, males bearing a YLE construct should have comparable fitness to wild-type males, since the persistence of suppression is directly related to this parameter[22]. Wild-type females mated with YLE[dsx]-A and YLE[dsx]-B males (n = 30) produced an equivalent number of larvae to females mated with wild-type males, hence corroborating that, under standard insectary conditions, the fertility of YLE[dsx]-A and YLE[dsx]-B males is not significantly different from the fertility of wild-type males from the same mosquito colony (Fig. 4b).

### The release of YLE[dsx] males induces prolonged suppression in caged mosquito populations

The main differentiating factor of the YLE technology, when compared to other self-limiting strategies, is the persistence of the suppression upon its release. To assess this in laboratory settings, we designed a single-release scenario experiment. We established an initial caged population of wild-type mosquitoes and introduced a number of YLE[dsx]; *dsxF*[Δ11/+] males (from the YLE[dsx]-B strain) equivalent to the initial number of wild-type males in the population (i.e. 300 wild-type females, 300 wild-type males and 300 YLE[dsx]; *dsxF*[Δ11/+] males). The experiment was performed in two replicate cages, and the populations in the next generations were seeded with 600 pupae randomly selected from the progeny of the previous generation. In every subsequent generation, we monitored the number of YLE[dsx] males and the number of wild-type females as an estimate of the population's reproductive and vectorial capacity.

In the first generation after the release, the number of wild-type females was reduced to 57% and 59% of the initial female population in cages 1 and 2, respectively (Fig. 5). Additionally, 49% and 43% of the males carried the YLE[dsx] construct in these populations. These results indicate that, in this experimental set up, YLE[dsx] males were able to efficiently compete with wild-type males for mating partners.

In an ideal scenario, the level of suppression would stay constant, and the YLE[dsx] males would remain at the released frequency. However, in the following generations, the number of wild-type females steadily increased to reach 86% and 87% in the fifth generation in the two replicate cages, while the frequency of YLE[dsx] males decreased to 27% and 14% of the total male population. The observed dynamics suggest a potential fitness cost associated to YLE[dsx] males, perceptible in this multigenerational experimental setup but that was not detected during the fertility assay performed on this strain.

To explore the dynamics of various alleles at the *dsx* locus in the caged populations, adults from each generation (from the first to the third generation post-release) were pooled to conduct amplicon sequencing of the locus. In the generation following the release, the *dsxF*[Δ11] allele was found at around 22% frequency in both cages (Supp. Fig. 6), which correlates with the inferred number of females with an intersex phenotype (43 and 41% of the total female population). By generation three post-release, the frequency of the *dsxF*[Δ11] allele decreased to 16% and 11% in cages 1 and 2, respectively, following the inverse tendency to the relative number of wild-type females. Of note, the sum of the frequencies of all other mutations in the target site remained continuously below 2% and was lower than 0.6% by the third generation post-release, indicating that the rise in the number of wild-

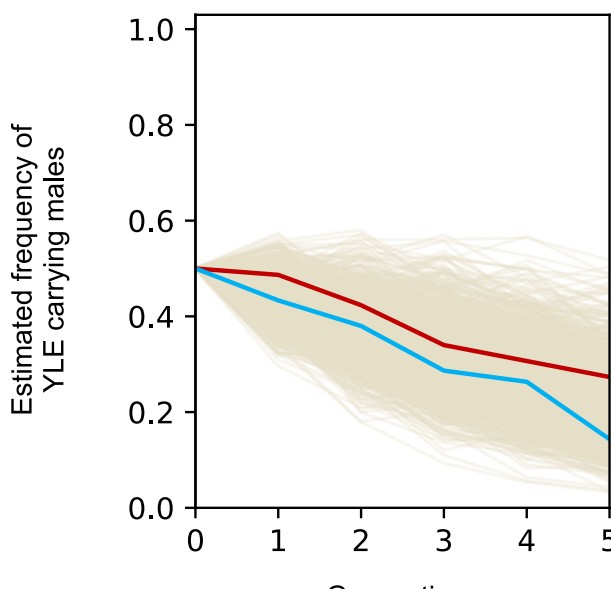

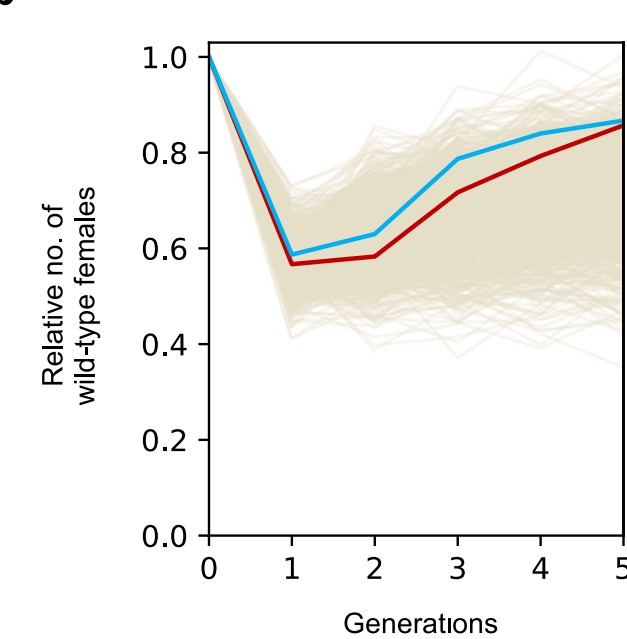

**Fig. 5 | Frequency of YLE[dsx] males and relative number of wild-type females across generations in caged populations. a** Estimated frequency of YLE[dsx]-carrying males through generations (assuming a total of 300 males in every generation post-release) in cage 1 (red) and in cage 2 (blue). An equal number of YLE[dsx] males (from the YLE[dsx]-B strain) and wild-type males were introduced in the release generation (generation 0). A constant decrease in the frequency of males carrying the YLE[dsx] construct was observed in both caged populations. Computer simulations of the cage trial are shown in cream (n = 1000). All simulations were done with parameters obtained from the phenotypic assays. YLE[dsx]-carrying males have a 22% reduced competitiveness compared to wild-type males, estimated by fitting the stochastic model to the experimental data using Approximate Bayesian Computation inference (see results on mathematical modelling). **b** Relative number of wild-type-looking females across generations in cage 1 (red) and in cage 2 (blue), used as a measure of the populations' reproductive and vectorial capacity. After a significant initial decline, the number of wild-type females steadily increased, reaching 86% and 87% of the initial size at generation five. Source data are provided as a Source Data file.

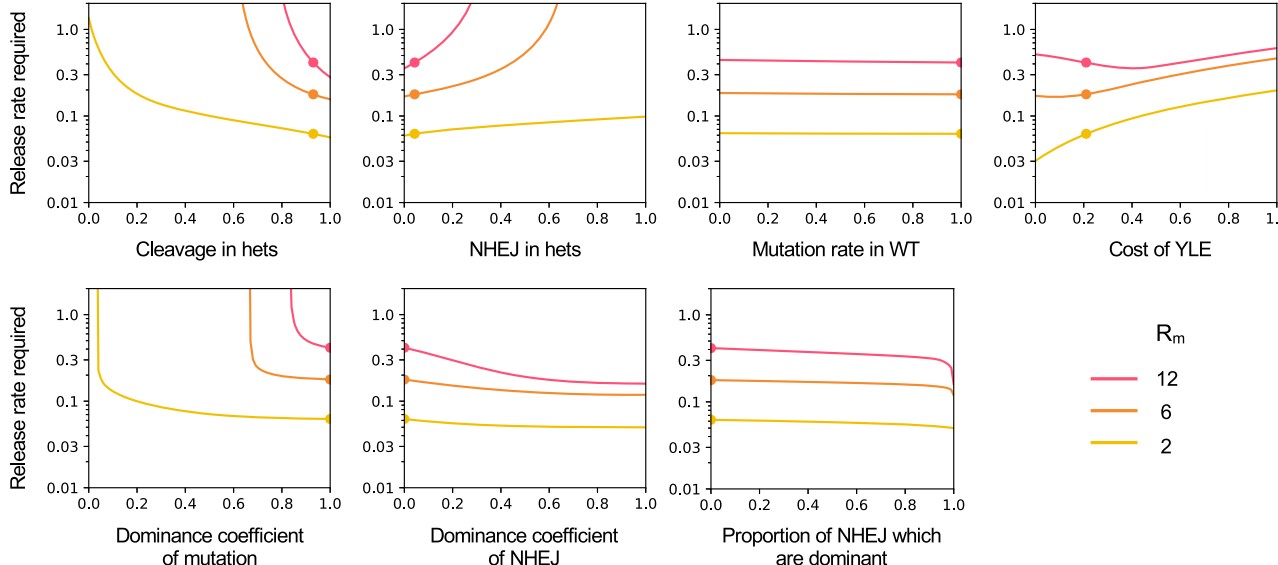

**Fig. 6 | Release rates required to achieve 95% suppression of a target population within 36 generations with males from the YLE^dsx-B strain as a function of various parameters, for $R_m$ = 12, 6, and 2 (red, orange, and yellow, respectively).** Lines illustrate the release rates required as a function of each parameter, where all other parameters are set to baseline values (Supp. Table 4). Dots indicate the current value of the developed YLE^dsx for each parameter. $R_m$ is defined as the intrinsic rate of increase of the population (expected number of daughters produced by a female in the absence of density-dependent mortality). Analysed parameters shown in the top row are (left to right): cleavage rate in YLE males heterozygous for a mutation (i.e., probability of cleaving the unmodified target in YLE males bearing one mutated copy), probability of generating a new mutation in the target by non-homologous end-joining (NHEJ) given cleavage in YLE males heterozygous for a mutation, probability of creating a mutation in the target site in YLE males that are wild-type homozygous, and fitness cost associated to YLE^dsx males. In the bottom row, analysed aspects include (left to right): the dominance coefficient of the released mutation (i.e., if the mutation carried by released YLE^dsx males is fully penetrant and always causes sterility in females, the dominance coefficient will be 1), the dominance coefficient of new dominant mutations generated by NHEJ, and the proportion of new mutations generated by NHEJ that are dominant.

**Table 1 | Comparison of the release rates required to suppress a target mosquito population with YLE^dsx males and other optimal self-limiting technologies**

| Suppression level (%) | YLE^dsx-B | Optimal YLE | Optimal SIT | Optimal RIDL | Optimal fs-RIDL | Optimal X-shredder | YLE^dsx-B; X-shredder (+/-) | YLE^dsx-B; X-shredder (+/+) |
|---|---|---|---|---|---|---|---|---|
| 95 | 0.180 | 0.037 [0.2] | 1.297 [7.2] | 0.438 [2.4] | 0.668 [3.7] | 0.396 [2.2] | 0.114 [0.6] | 0.088 [0.5] |
| 99 | 0.190 | 0.041 [0.2] | 1.300 [6.8] | 0.440 [2.3] | 0.678 [3.6] | 0.402 [2.1] | 0.122 [0.6] | 0.095 [0.5] |

Release rates required to achieve a suppression level of 95 and 99% in a target mosquito population with an intrinsic rate of increase ($R_m$) of 6, assuming releases for 36 generations. Release rates were calculated as a proportion of the initial number of males in the target population for the developed YLE^dsx-B strain, an optimal YLE causing female-specific sterility, an optimal SIT, an idealised RIDL (causing unspecific lethality after density-dependence), an optimal fs-RIDL (causing female-specific lethality after density-dependence), an idealised autosomal X-shredder, and males from the YLE^dsx-B strain bearing an autosomal X-shredder construct in heterozygosity (+/-) or homozygosity (+/+). For the optimal autosomal approaches, the release of homozygous males is assumed. Numbers in square brackets indicate how many times larger the release rate would have to be compared to the release of males from the YLE^dsx-B strain to achieve the same level of suppression.

type females was not due to the emergence and accumulation of recessive mutations in the target (Supp. Fig. 6).

### Mathematical modelling suggests a great potential for the YLE as a self-limiting genetic control tool

To assess the potential of the best-performing YLE^dsx-B strain as a tool to suppress a target mosquito population in the wild, we used the life history traits observed in laboratory conditions for this strain as parameters in a mathematical model.

First, to infer the potential fitness cost in YLE^dsx males, we simulated a single release of YLE^dsx; $dsxF^{AII/+}$ males in caged populations using a stochastic model and compared the predictions to the empirical results. The outcome suggested a 22% fitness reduction in YLE^dsx males compared to wild-type ones (Supp. Fig. 7, Figs. 5, 6). Still, two parameters are unknown in this model: 1) the rate of de novo mutations in that fraction of YLE^dsx males that, in subsequent generations post-release, may have two unmodified $dsx$ alleles; and 2) the landscape of other mutations in that locus that cause an intersex phenotype, and their penetrance. For these reasons, a conservative approach was taken assuming that every male that did not inherit the released dominant mutation would generate a recessive one that

would be transmitted to their progeny at the same frequency as the dominant mutation in released males.

We then used a deterministic model to determine the release rates required to achieve 95% and 99% suppression in 36 generations for a target mosquito population with an intrinsic rate of increase of $R_m$ = 6, assuming equal-sized releases in every generation. The model showed that the developed YLE^dsx would require seven times lower releases than an optimal SIT strategy, two to four times lower releases than optimal RIDL or fs-RIDL technologies, and two times lower releases than an idealised X-shredder (Table 1). Nevertheless, our YLE^dsx strain would still need four to five times larger releases than an optimal YLE.

Original mathematical modelling on this strategy showed that the release rates could be reduced if YLE males also carry an autosomal X-shredder construct that leads to a disproportionate transmission of the Y chromosome[22]. Correspondingly, our model with the YLE^dsx strain parameters shows that its combination with an autosomal X-shredder would halve the release rates required to achieve the same levels of suppression (Table 1).

To understand the source of the difference between our YLE^dsx strain and an optimal YLE, we performed a sensitivity analysis of how the release rate required varies as a function of changes in the

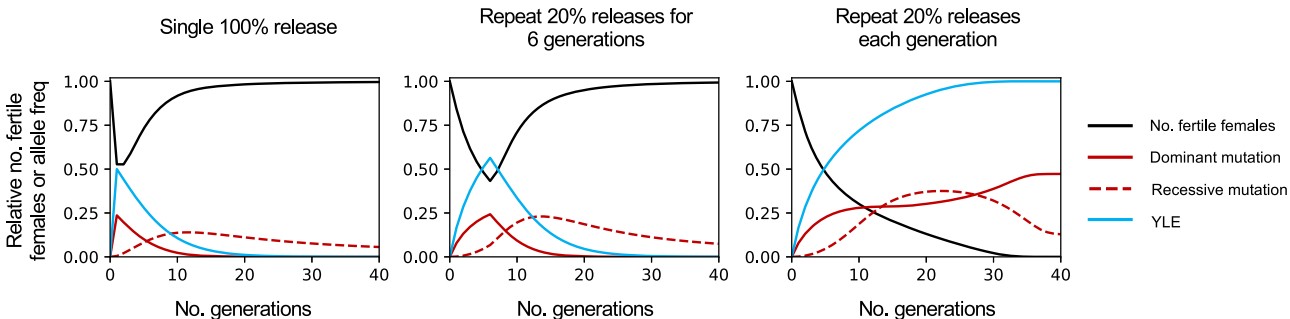

**Fig. 7 | Time course of a population suppression upon releases of YLE^dsx males heterozygous for the dominant *dsxF^Δ11* allele.** The black line shows the relative number of fertile females in a target population across generations following a single release of YLE^dsx; *dsxF^Δ11/+* males equivalent to the initial number of males in the population (*left*), and releases of YLE^dsx; *dsxF^Δ11/+* males equivalent to 20% of the initial male population for 6 generations (*middle*) or indefinitely (*right*). The

frequency of males bearing the YLE construct is represented in blue. The frequency of the released dominant *dsxF^Δ11* mutation in the population is shown by the continuous red line, and the dashed red line indicates the frequency of emerging recessive mutations. All simulations were done with parameters obtained from the phenotypic assays (including a 22% reduced competitiveness of YLE^dsx-carrying males compared to wild-type males) and assuming $R_m = 6$.

underlying genetic variables (Fig. 6). Among all the analysed variables, it is the rare production of recessive alleles, and the fitness cost associated to the YLE^dsx that are the most critical, while the remaining parameters are very close to the optimum values.

Recessive mutations produced by NHEJ repair mechanisms would accumulate in the target population because they could not be recognized and cleaved by the Cas9-gRNA complex and they would not be selected out as efficiently as dominant mutations, since females that are heterozygous for these alleles would be fertile and would contribute to their transmission. Nonetheless, while the YLE^dsx fitness cost implies a declining level of suppression after halting releases, neither this nor the accumulation of recessive mutations would prevent the complete suppression of a target mosquito population following repeated releases of YLE^dsx; *dsxF^Δ11/+* males (Fig. 7).

## Discussion

In this study, we used the *vasa* regulatory sequences to drive the expression of a Y-linked Cas9 cassette in the germline of *An. gambiae*. We show high activity of the endonuclease that was able to promote efficient homing of a female-specific dominant mutation (*dsxF^Δ11*) that we identified at the *dsx* locus, leading to a high proportion of females inheriting a copy of this mutation and being fully sterile. The differences observed at two distinct Y chromosome integration sites, in terms of overall Cas9 activity and variability in this activity among males with the same engineered Y chromosome, seem to indicate that this chromosome is affected by position-effect variegation through its length. This is particularly relevant for the development of future genetic control strategies linked to the Y chromosome in *An. gambiae*, and efforts to locate the relative positions of the two Y chromosome integrations described in this study could provide useful information for the identification of chromosome regions that are more transcriptionally active and less prone to silencing.

The multigenerational experiment showed that a single release of YLE^dsx males could effectively decrease the size of the targeted mosquito population and have a long-lasting suppressive impact. However, the estimated 22% decrease in fitness of YLE^dsx males means that the initial level of suppression cannot be maintained indefinitely. This is because the frequency of these males gradually declines, allowing the population to slowly rebound in the generations following the release. Nonetheless, the described model demonstrates that repeated releases of YLE^dsx; *dsxF^Δ11/+* males could lead to the collapse of the target population.

The fitness effect on YLE^dsx males observed in the multigenerational experiment was not detected in the fertility assay performed on this strain, highlighting the utility of the former in making more accurate predictions of performance. The apparent fitness

costs may be explained by an impairment of these individuals in development, survival, or mating competitiveness. Alternatively, perhaps the *dsxF^Δ11* allele has subtle effects on male fitness. Regardless of its origin, the fitness cost associated with the YLE^dsx males may have benefits from a regulatory and environmental standpoint. By releasing these males, a target population could be suppressed to a specific level for a sufficient period of time, theoretically resulting in a positive epidemiological impact. Once the releases are stopped, the transgene would ultimately disappear from the population rather than persisting indefinitely, as would be the case in the absence of fitness costs. Moreover, the cost also means the transgene will spread less to neighbouring populations. This technology's sustained effect means that releases can occur every other generation and still maintain a certain level of suppression, offering a significant logistical advantage.

In addition, while the release rates required for YLE^dsx-B are four to five times larger than for the optimal YLE, these are still significantly lower than the release rates for other optimal self-limiting strategies. Given the cost and logistic challenges associated with the mass-rearing of mosquitoes, the use of a YLE could provide substantial operational and resource efficiencies over current approaches to field releases of vector control agents. Furthermore, combining the YLE^dsx with an autosomal X-shredder would halve the required release rates.

While sex-specific dominant mutations have been widely studied in the *doublesex* gene of *D. melanogaster*[34–36,38,39], none had been previously described in *An. gambiae* mosquitoes. The molecular basis of the dominant phenotype may be interference of the truncated form in proper functioning of the homodimer that is necessary for DSX function ('dimer poisoning')[40] (Supp. Fig. 8) or changes in the splicing patterns of the male or female isoform[35,36,38,39]. Regardless of the precise mechanism, the *dsxF^Δ11* allele may not be unique in giving a dominant phenotype as we also found a one-base pair deletion with apparently the same effect (Supp. Figures 2, 5, Supp. Table 3).

According to our modelling, the ratio of novel dominant negative *versus* a recessive loss of function mutations in the target gene within the offspring that do not receive a homed allele will significantly impact the efficiency of the YLE strategy. Ideally, 100% homing rates of the *dsxF^Δ11* allele would solve this issue; however, this is unlikely to be achieved. To address this, target genes for this type of YLE should not easily tolerate modifications that include recessive mutations or alleles resistant to cleavage that restore the gene function. Although *dsx* is a functionally constrained gene, it would be worthwhile to explore the potential for genetic variation within the female-specific exon and its surroundings to generate dominant mutations over recessive or functional resistant ones. As with CRISPR-based gene drives, using multiple gRNAs would be expected to significantly retard

resistance[5,37,41–43]. Alternatively, two genes that induce a female-specific dominant effect could be targeted at the same time. Additional targets could be explored by looking for genes whose proteins form oligomers and have an essential role only in females, such as the *Act4* gene in *Aedes aegypti* and *Culex quinquefasciatus* mosquitoes, for which different in-frame deletions cause a dominant flightless phenotype in females[44,45]. Haplo-insufficient genes on the X chromosome might also be used as targets, as long as they are not required in male spermatogenesis, though this would require recurrent editing rather than homing[46,47].

With regards to preparatory work for its application, a YLE strain would be easily maintained and scaled up prior to field releases. In an insectary facility, male larvae can be automatically sorted due to the Y chromosome-linked fluorescent marker, and then crossed to wild-type females. If the aim is to release the YLE paired with a dominant mutation such as *dsxF^AII*, given that it is not possible to generate a homozygous strain, it might be desirable to link this allele to a fluorescent marker in a way that it does not affect the dominant phenotype (e.g., by placing the marker in a nearby intron), and therefore ensure that all the released males bear this mutation. High-throughput larval screening using flow cytometry[48] could allow the effective selection of larvae containing the YLE construct and the dominant mutation(s).

The YLE^dsx strain presented here is a self-limiting genetic control tool for *An. gambiae* that would seem to be more effective than other self-limiting vector control technologies: it has a stronger effect on population suppression that is maintained for a longer period of time. At the same time, it is less invasive than self-sustaining approaches and it is expected to be more geographically confined. This YLE technology expands the toolbox of genetic control strategies and offers a middle-ground option between existing self-limiting technologies and self-sustaining ones.

## Methods

### Plasmids used for the generation of the YLE strains

The plasmid p16510, which had been previously cloned[8], contains a human-codon-optimized Cas9 coding sequence under the control of the *vas2* promoter[31], a gRNA targeting the intron 4 – exon 5 boundary of the *dsx* gene under the ubiquitous *U6* promoter (gRNA spacer sequence: 'GTTTAACACAGGTCAAGCGG'), and a gene coding for RFP under the *3xP3* promoter. Additionally, a modification of this plasmid existed with the sole difference of a gene coding for CFP instead of RFP.

### Generation of the YLE strains targeting *dsx*

To generate the YLE^dsx-A strain, embryos from the Y-AttP line[23] were injected with a solution containing the p16510-CFP plasmid (at 200 ng/µL) and a *vasa-integrase* helper plasmid (at 400 ng/µL) for integrase-mediated insertion. Microinjections were performed as described previously[49]. In order to create the YLE^dsx-B strain, we injected embryos from the Ygfp strain[32] provided by Jaroslaw Krzywinski (bearing a transgene on the Y chromosome with a GFP-coding gene flanked by two attP sites for integrase-mediated recombination) with solution containing the p16510-RFP plasmid (at 200 ng/µL) and a *vasa-integrase* helper plasmid (at 400 ng/µL).

Surviving L1 larvae were screened for transient expression of the fluorescent marker in the anal papillae and parts of the ventral nerve chord. Larvae with and without this transient expression were reared separately and crossed to wild-type mosquitoes at adulthood. The progeny from these crosses was screened for expression of the fluorescent marker under the *3xP3* promoter, visible in the optic lobe of the head and the dorsal ganglia across the larval body. Successful integration events were detected by the exchange of the fluorescent marker in G1 male individuals that were RFP⁻ and CFP⁺ for YLE^dsx-A, or GFP⁻ and RFP⁺ for YLE^dsx-A.

### Assessment of homing levels of the *dsxF^GFP-null* allele

To ensure that each progeny analysed for homing is coming from a different male, the experimental crosses were performed as described by Rogers et al.[50] and explained below.

YLE^dsx males were initially mated with females heterozygous for the *dsxF^GFP-null* allele. Individuals in the progeny with expression of the marker linked to the YLE and GFP (linked to *dsxF^GFP-null*) were selected. To ensure that YLE^dsx males used in the experimental cross would not have inherited other mutations in the target site that would prevent homing of *dsxF^GFP-null*, the selected males were mated with wild-type females. Around 600 double-positive individuals were selected in the offspring at the pupal stage and placed in a large cage. 100 females from the *An. gambiae* wild-type colony were sexed at the pupal stage and placed in a separate cage. Six days after emergence, at the time of the programmed dusk, females were introduced into the male cage in groups of 6-8. Mating pairs falling to the bottom of the cage were isolated by gently placing a 25 mL plastic glass over them until copulation terminated. Once the couple separated, the plastic glass was brought over a thin piece of cardboard previously left on the bottom of the cage and the mated mosquitoes were transferred to another cage. Mated females were kept in this separate cage and males were removed to avoid a new mating with another female. 50-60 mated females were captured. The following day to the mating, females were fed with blood and 3 days later they were allowed to lay eggs individually. Hatched L1 larvae were screened for the presence of GFP. Larvae from progenies with low homing rates of *dsxF^GFP-null* were pooled to perform genomic DNA extraction (Wizard® Genomic DNA Purification Kit, Promega, A1125) and subsequently do amplicon sequencing (GENEWIZ) on the *dsx* target to check for the presence of mutations that could have prevented the homing (primers used: <u>ACACTCTTTCCCTACACGACGCTCTTCCGATCT</u>ACTTATCGGCATCAGTTGCG and <u>GACTGGAGTTCAGACGTGTGCT</u><u>CTTCCGATCT</u>GTGAATTCCGTCAGCCAGCA, Illumina adapters for amplicon sequencing are underlined). Amplicon sequencing data were analysed using CRISPResso2[51].

While the procedure to obtain YLE^dsx; *dsxF^GFP-null/+* males was the same for both YLE^dsx lines, the YLE^dsx-B line was screened by eye under the fluorescence microscope, but the YLE^dsx-A line was screened using the COPAS system to unambiguously detect males expressing GFP. Homing of the *dsxF^GFP-null* allele in the YLE^dsx-A line was assessed by eye under the fluorescence microscope and evaluated only in females to avoid confusion with CFP⁺ only males.

Statistical differences in homing rates between both strains were assessed using a Mann-Whitney test.

### Assessment of homing levels of the *dsxF^AII* allele

To ensure that the YLE^dsx males used for the experimental cross had the markerless *dsxF^AII* allele, the offspring of different females from these lines were reared separately. Individuals were sexed at pupal stage, looking for progenies where all the females showed the intersex phenotype (indicating that all of them had the dominant mutation, and therefore their sibling males did too). The males of those progenies were selected to perform the experiment: 50 YLE^dsx males supposedly bearing the dominant mutation were allowed to mate for 5 days to 60 wild-type females in a cage. Females were then fed blood and allowed to lay eggs individually 3 days later. Progenies were raised separately and sexed at pupal stage. Females from each progeny were then pooled to perform genomic DNA extraction (Wizard® Genomic DNA Purification Kit, Promega, A1125) and subsequently do amplicon sequencing (GENEWIZ) on the *dsx* target to check the frequency of the *dsxF^AII* allele in the offspring. Amplicon sequencing data were analysed using CRISPResso2[51].

### Fertility assays

YLE^dsx males from both lines were crossed with wild-type females in groups of 50. They were left to mate for 5-7 days before being blood-

fed. A minimum of 30 females were allowed to lay eggs individually by placing them in separate cups 3 days after blood feeding. After 2 nights, the number of eggs laid by each female was recorded. The number of larvae hatching from each progeny in the following days was recorded as well.

Females with an intersex phenotype were crossed with wild-type males in groups of 60. They were left to mate for 5-7 days before being blood fed. A minimum of 50 females were allowed to lay eggs individually by placing them in separate cups 3 days after blood feeding. After 2 nights, the number of eggs laid by each female was recorded. The number of larvae hatching from each progeny in the following days was recorded as well. The same procedure was followed with wild-type females as a control.

Statistical differences against wild-type reference crosses were assessed using a Kruskal-Wallis test.

### Blood-feeding ability assay

75 individuals with an intersex phenotype (from the progeny of YLE$^{dsx}$ males) and 50 wild-type females were collected at the pupal stage and separated into cages of 25 individuals for easier tracking. 4-5 days after emerging as adults, glucose was removed from the cage 45 minutes before offering a blood meal for 30 minutes. Individuals attempting to feed 5 minutes after the blood meal was offered were categorised as 'attracted to blood', and individuals that had fed on blood by the end of the 30 minutes were categorised as 'able to feed on blood'. Statistical differences were evaluated performing a Fisher's test on the number of individuals in 2×2 contingency tables for each of the outcomes observed (attraction / ability to feed).

### Dissections of females showing an intersex phenotype

Adult females with an intersex phenotype were anaesthetised on ice for 5–10 minutes. Individuals were then place on a glass slide with PBS solution and were dissected under a stereomicroscope using two needles. One needle was used to hold the thorax of the females while the other one was placed on the last segment of the abdomen to pull. Gonads and spermathecae were isolated from the rest of the body for visualization and recording under an EVOS XL Core Cell imaging system. Dissections of wild-type males and females were also performed for comparison.

### Experimental cross with the *R1* allele strain

25 YLE$^{dsx}$ adult males were crossed with 25 females homozygous for the *R1* allele. They were allowed to mate for 5-7 days before being blood fed, and larval progeny was kept in trays. Individuals were sexed at pupal stage and separated according to their phenotype (intersex or wild-type-looking female) for pooled genomic DNA extraction (Wizard® Genomic DNA Purification Kit, Promega, A1125) and subsequent amplicon sequencing of the *dsx* target (GENEWIZ).

The same procedure was followed after 50 YLE$^{dsx}$ males bearing the *R1* and the *dsxF^{AII}* alleles were mated to 50 wild-type females.

### Cage trial experiments

Two cages were set with an initial population of 300 wild-type females, 300 wild-type males and 300 YLE$^{dsx}$ males. The wild-type and the YLE$^{dsx}$ lines were reared in parallel under the same conditions, and age-matched individuals were added at adult stage in this first generation (G0). Mosquitoes were left to mate for 6-7 days prior to providing a blood meal. Three days later, females were allowed to lay eggs in a 100 mL glass bowl with 90 mL of salted water and a cone-shaped filter paper covering it. Eggs were carefully sprayed into plastic trays with 500 mL of water and filter paper on the sides to prevent the eggs from sticking to the walls of the trays and drying. Hatched L1 larvae were split into eight trays of 200 individuals each. 600 randomly selected pupae were screened for the presence of the YLE$^{dsx}$-linked RFP marker

and sexed to score the number of wild-type-looking females, and they were then allowed to emerge together in a cage.

### Mathematical modelling

To model the impact of the YLE$^{dsx}$ for population suppression we developed two types of models, a deterministic model used to explore the impact of releasing YLE males into a wild population and a stochastic model used to simulate the empirical cage trials and estimate the unknown fitness costs associated with the YLE using Approximate Bayesian Computation. The impacts of the YLE on gamete inheritance and fitness were modelled in the same way for both models; however, they differed in their population dynamics. In short, we model a Y-linked locus with two alleles: the wild-type ($Y$) and the YLE ($y$); and an autosomal target locus with three alleles: the wild-type ($A$), a cleavage-resistant recessive mutation ($a$) and dominant mutation ($\alpha$). No functional cleavage-resistant alleles were considered. Therefore, considering both the Y-linked and autosomal loci, there are a total of 12 male genotypes and 6 female genotypes. We extend the deterministic model to also simulate the YLE in the presence of an autosomal sex distorter construct which results in cleavage of the X chromosome during spermatogenesis, increasing the proportion of Y-bearing sperm (X-shredder). We include a second unlinked autosomal locus with two alleles, the wild-type ($B$) and the X-shredder construct ($b$). For the extended model, considering the Y-linked and both autosomal loci, there are a total of 90 male genotypes and 45 female genotypes. Using these models, it is also possible to simulate a range of comparative strategies including release of sterile males (SIT), release of individuals carrying a mutation which causes death after density dependent mortality in both sexes (RIDL) or only in females (fsRIDL) and an X-shredder alone. All simulations were performed in Julia, a scientific programming language[52] within the Jupyter notebook interface[53]. Code used for the simulations is available on GitHub (https://github.com/KatieWillis/YLESimulator)[54] and further information can be found in the Supplementary Methods.

### Statistics and reproducibility

The statistical tests and sample sizes used in each experiment are described in the corresponding methods section. No statistical method was used to predetermine sample size, which was chosen consistent with the previous literature reporting similar assays. Sample size was maximised within the feasibility of performing biological assays with live insects. No data were excluded from the analyses.

### Reporting summary

Further information on research design is available in the Nature Portfolio Reporting Summary linked to this article.

## Data availability

Amplicon sequencing data generated and analysed in this study are available in the Sequence Read Archive (SRA) with the accession code PRJNA1076121. Source data are provided with this paper.

## Code availability

Code used for the simulations performed in this study is fully available on GitHub: https://github.com/KatieWillis/YLESimulator, under a GNU General Public License 3.0 (GPL-3.0).

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

## Acknowledgements

We would like to thank Jaroslaw Krzywinski and Elzbieta Krzywinska for providing one of the *An. gambiae* Y docking-site strains used in this work, and Manolo Fernandez Perez for advising on the parameter inference. We also thank John Connolly and the Target Malaria Regulatory Team for their valuable feedback. This work was supported, in whole or in part, by the Bill & Melinda Gates Foundation [Grant Number INV006610 "Target Malaria Phase II"] and Open Philanthropy (OPP1210755) (A.B., A.C.). Under the grant conditions of the Foundation, a Creative Commons Attribution 4.0 Generic License has already been assigned to the Author Accepted Manuscript version that might arise from this submission.

## Author contributions

T.N. and A.C. conceived the project; I.T., T.N. and F.B. designed the research; I.T., M.G. and L.P. performed the experiments; I.T., T.N. and F.B. analysed data; K.W. and A.B. performed mathematical modelling; A.B. and A.C. acquired funding; and I.T., T.N. and F.B. wrote the paper with input from all authors.

## Competing interests

A.C. is a founder of Biocentis, Ltd. A.C. and T.N. have equity interest in Biocentis, Ltd. The remaining authors declare no competing interests.
