## [Peer Review file · Nature Communications]

A Y chromosome-linked genome editor for efficient population suppression in the malaria vector *Anopheles gambiae*

Corresponding Author: Dr Federica Bernardini

Version 0:

Reviewer comments:

Reviewer #1

(Remarks to the Author)

Summary: The manuscript by Tolosana et al., describes an innovative and promising new sterile insect technology (SIT) wherein a dominant female sterility phenotype is induced by a CRISPR gene cassette (YLE) located on the Y-chromosome. The concept is very elegant. A male carrying the YLE construct induces copying of the dominant female sterile allele to the homologous chromosome during meiosis such that nearly all female progeny (>90%) resulting from a YLE male mating with a wild-type female inherit that dominantly acting allele, and are thus rendered sterile. A key feature of this new system is that the YLE element is expected to persist in a target population in contrast to standard SIT systems that typically disappear after only one or two generations. Mathematical modeling shows that this agile long-acting genetic YLE system out-performs a variety of existing short-acting SIT systems. This is an impressive claim worthy of further study.

Critique and recommendation: Experiments presented in this manuscript are well performed and support the authors' key claims. In my view, this study merits publication in Nature Communications. The authors may wish to consider two major points in preparing a final version of their manuscript.

Major points:

1) The authors conclude that the dsx F11 frame-shift mutation is acting as a dominant negative allele based on sequencing female mosquitoes that either display or not the sterilizing intersex phenotype. They find that pools of intersex-scored females carry the deletion, but that this mutation is only rarely found in pools of wild-type-scored females (note: exactly how this scoring is done is not entirely clear from either the text or methods sections - it would be helpful to know for example whether all intersex pools carried the deletion and whether it constituted 50% the reads in such pools). The authors speculate reasonably that the dominant negative phenotype arises from formation of inactive Dsx protein dimers (as has also been reported in *Drosophila*) and also show alpha-fold modeling supporting the plausibility of this hypothesis. Although this explanation may well be correct, it also seems that there is an alternative possibility, namely that leaky expression of vasa-Cas9/gRNA transgenes leads to a mosaic inactivation of the second allele in these females which may (most of the time) or may not (rarely) be sufficient to produce an effective homozygous recessive dsx- mutant phenotype. One way the authors might answer this question is to see whether they can separate the dsx F11 allele from the vasa-cas9 transgene on the Y chromosome using the rare wild-type appearing females carrying the dsx F11 allele. Such dsx F11/+ females could then be crossed to wild-type males, and assessed for whether they again manifest the intersex phenotype. In the absence of such a test, perhaps this point could be raised in the discussion as an alternative possible explanation for the dominant behavior of the dsx F11 allele?

2) The only obvious experiment missing from the current study is an actual multigenerational cage test to show that the YLE element performs as expected according to the modeling. Although I do not recommend requiring inclusion of this experiment for publication, I do think it would greatly increase the impact of the study if it were added. If the authors have already initiated such experiments, perhaps they might include them in the final version of the manuscript (even if only a few generations are available at this time)? In particular, it would be very interesting to compare the experimental performance of the YLE to another standardly employed transiently acting genetic SIT system.

Minor points:

1) It is not exactly clear how the authors derived the dsx F11 allele and whether it might have been isolated from the first set of experiments using the GFP-null dsx allele. Was the dsx F11 allele obtained as an NHEJ induced copying error from those crosses?

2) The authors mention in the discussion that an important future issue to assess would be to estimate the rate at which functional NHEJ alleles might be generated by the YLE system. If they have any data pertaining to this important question it would be good to include it. For example, it would be good to know what fraction of NHEJ alleles they have sequenced are in-frame. It might also be good to discuss possible work arounds (in addition to those they already outline) in which the YLE element also expresses a second gRNA that targets a highly conserved dsx splice site (the same site that is targeted in the highly efficient gene-drives this same group has designed). Expressing this splice-site gRNA should generate a parallel recessive loss-of-function allele (no functional NHEJ alleles have observed to be generated at this site). This dual strategy might guarantee that all progeny were at least recessive sterile?

(Remarks on code availability)

Reviewer #2

(Remarks to the Author)

In this manuscript, Tolosana et al describe a Y-linked gene editing system capable of reducing the reproductive potential of *Anopheles gambiae* mosquitos. This follows closely from their previous work developing Cas9-based gene drive mosquitoes targeting the female-specific form of the doublesex gene. New components described in this paper are the source of Cas9, which has been moved out of the dsx locus (Kyrou et al 2018) and onto the Y-chromosome, and the development of a dominant negative deletion in dsx that induces intersex phenotypes. This contrasts with their previous disruption of dsx (insertion of EGFP), where heterozygous individuals formed normally, and only homozygous disruptions were intersex. Using these tools, the authors demonstrate that in the presence of paternally-supplied Cas9, conversion of WT dsx alleles to either EGFP or deletion alleles via gene drive is efficient. Modeling frameworks compare the efficiency of the approach to other self-limiting approaches, and suggest comparable or superior results. Overall, this is an important finding and would be of general interest, but several aspects of the manuscript are underdeveloped.

The authors utilize two Y-linked strains that exhibit different levels of gene drive, which the authors attribute to variable expression based on position. However, the authors do not attempt to measure expression levels or determine the position of the insertions.

The authors describe a second deletion (1bp) as also being potentially dominant negative, but make no attempt at discerning what aspects of these mutations lead to DN phenotypes. Do these maintain female-splicing but disrupt the ORF? Or alternatively, do these alone disrupt female splicing, and increase usage of the male form in XX individuals (leading to intersex)? As the strategy described by the authors depends on these phenotypes, as does the frequency by which new DNs arise, a basic understanding of this matter is highly relevant to the conclusions.

The authors limit their data collection to single female one generation outputs. It is somewhat surprising that they did not perform multi-generational studies as they have done in so many past studies given the model predictions that population-level effects should be rapid (10 generations for 80% reduction). If there are technical reasons as to why this is not possible/practical, what does this mean for the maturity of this approach?

Modelling assumes no fitness costs for YLE males, with only the fertility values demonstrated experimentally. While I understand it is not possible to perform every possible fitness test, I am surprised the modelling did not examine the tolerance of the approach for fitness costs in males (lifespan/mating competitiveness/developmental timing)

Figure 2: Panel (a) shows a construct, but not how it is targeted to the Y chromosome. The details are in the methods, but it maybe more clear to indicate on the figure. Panel (c) refers to "modified dsx allele" and is colored in red, but really this is a transgenic EGFP allele. Showing this directly and labelling it as "dsxF-EGFPnull" would match the labels in panel (d) and make the figure much easier to follow. Panel (d) refers to colors (yellow, pink) that differ from the colors in the figure (maroon, red). Some symbols maybe did not convert correctly on the pdf (81.5 box 3.2 should be 81.5 +/- 3.2?) Dashed line is referred to in the legend but is not visible in the figure.

Figure 3. Dominant negative females do not lay eggs, but do they feed? This is never stated, and contrasts with their previous paper (Kyrou et al 2018) where homozygous dsx females were offered blood but did not feed. If they do feed, they could still be vectors even if sterile, so this is important to mention.

Figure 4. dsxF11 allele and male YLE fertility. Symbols corrupted again. Colors referenced (yellow, pink) do not correspond to colors in figure.

Line 128: "dictated" is a strange term here. "facilitated", "was advantageous" or something similar may be more accurate. Nobody forced you to choose dsx.

Line 135-136: give reference for docking sites generated on Y chromosome. (lines 264-265: positions unknown?). I see the

reference in the methods (23) for Line A , but Line B has no reference. Since that was the strain that performed the best, this is even more important to provide information on this site and the construct/method used to integrate it. Are the docking strains and YLE strains available for others to use? How can they be shared?

Line 141-143: why not measure Cas9 expression directly (mRNA or protein) to validate proposed expression differences?

(Remarks on code availability)

Version 1:

Reviewer comments:

Reviewer #1

(Remarks to the Author)

The authors have addressed all of my minor prior concerns. I recommend publication of this interesting and impactful manuscript.

(Remarks on code availability)

Reviewer #2

(Remarks to the Author)

The authors had productively addressed the critical aspects of my comments. Despite the multi-generational study producing some unexpected outcomes, the dominant negative dsx phenotype is an important advance and will no doubt inspire additional work looking both to adapt this approach and to improve it. congratulations to the authors for putting together such a strong study.

(Remarks on code availability)

POINT-BY-POINT RESPONSE TO THE REVIEWERS' COMMENTS

Reviewer #1 (Remarks to the Author):

Summary: The manuscript by Tolosana et al., describes an innovative and promising new sterile insect technology (SIT) wherein a dominant female sterility phenotype is induced by a CRISPR gene cassette (YLE) located on the Y- chromosome. The concept is very elegant. A male carrying the YLE construct induces copying of the dominant female sterile allele to the homologous chromosome during meiosis such that nearly all female progeny (>90%) resulting from a YLE male mating with a wild-type female inherit that dominantly acting allele, and are thus rendered sterile. A key feature of this new system is that the YLE element is expected to persist in a target population in contrast to standard SIT systems that typically disappear after only one or two generations. Mathematical modeling shows that this agile long-acting genetic YLE system out-performs a variety of existing short-acting SIT systems. This is an impressive claim worthy of further study.

Critique and recommendation: Experiments presented in this manuscript are well performed and support the authors' key claims. In my view, this study merits publication in Nature Communications. The authors may wish to consider two major points in preparing a final version of their manuscript.

Major points:

1) The authors conclude that the dsxDF11 frame-shift mutation is acting as a dominant negative allele based on sequencing female mosquitoes that either display or not the sterilizing intersex phenotype. They find that pools of intersex-scored females carry the deletion, but that this mutation is only rarely found in pools of wild-type-scored females (note: exactly how this scoring is done is not entirely clear from either the text or methods sections - it would be helpful to know for example whether all intersex pools carried the deletion and whether it constituted 50% the reads in such pools). The authors speculate reasonably that the dominant negative phenotype arises from formation of inactive Dsx protein dimers (as has also been reported in *Drosophila*) and also show alpha-fold modeling supporting the plausibility of this hypothesis. Although

this explanation may well be correct, it also seems that there is an alternative possibility, namely that leaky expression of vasa-Cas9/gRNA transgenes leads to a mosaic inactivation of the second allele in these females which may (most of the time) or may not (rarely) be sufficient to produce an effective homozygous recessive *dsx*-mutant phenotype. One way the authors might answer this question is to see whether they can separate the *dsx*DF11 allele from the vasa-cas9 transgene on the Y chromosome using the rare wild-type appearing females carrying the *dsx*DF11 allele. Such *dsx*DF11/+ females could then be crossed to wild-type males, and assessed for whether they again manifest the intersex phenotype. In the absence of such a test, perhaps this point could be raised in the discussion as an alternative possible explanation for the dominant behavior of the *dsx*DF11 allele?

We thank the reviewer for this useful comment. Concerning the sequencing data obtained for pools of intersex-scored females and wild-type-scored females, we acknowledge that the results were not accurately described. In response to the reviewer's concerns, we have rewritten this section, emphasising that all intersex-scored females showed 50% of the reads containing the 11-base pair deletion along with 50% of the reads containing the wild-type allele (lines 186-192), which would argue against the hypothesis of the paternally deposited nuclease as the cause of the intersex phenotype. We added the Supplementary Figure 1 showing the frequencies of the wild-type and the *dsx*^{Δ11} alleles in the pool.

Regarding the female dominant sterility caused by the 11-base pair deletion, we wanted to further address the possibility, suggested by the reviewer, of leaky expression of Cas9 in females as an alternative explanation for the dominant intersex phenotype. As females do not carry the Y chromosome, Cas9 will not be present in the genome of these individuals. However, another possibility, still in line with the reviewer's concerns, would be the paternal deposition of Cas9. To address this, we have conducted an experiment in which female individuals were generated with a genetic background carrying the *dsx*^{Δ11} allele in conjunction with a single SNP within the *dsx* target sequence that is functional and resistant to Cas9 cleavage (R1 allele). Our results showed that in these females, the dominant intersex phenotype persisted, ruling out the possibility of additional recessive mutations in the maternally inherited allele that paternally deposited Cas9 might have generated. This experiment has now been included in the 'results' paragraph describing the 11-base pair deletion causing

female dominant sterility, making our speculation much more solid (lines 196-216, Figure 3b, c, Supplementary Figure 2).

2) The only obvious experiment missing from the current study is an actual multigenerational cage test to show that the YLE element performs as expected according to the modeling. Although I do not recommend requiring inclusion of this experiment for publication, I do think it would greatly increase the impact of the study if it were added. If the authors have already initiated such experiments, perhaps they might include them in the final version of the manuscript (even if only a few generations are available at this time)? In particular, it would be very interesting to compare the experimental performance of the YLE to another standardly employed transiently acting genetic SIT system.

We appreciate the suggestion of a multigenerational experiment. Our initial intention was to show proof of principle for the approach and the building of its components, which represent a step change in bringing more powerful versions of self-limiting genetic control strategies. However, we agree with the reviewer that cage experiments would add to the impact of the paper. We have now performed these and the results show that a single release of YLE^{dsx} males could effectively decrease the size of the targeted mosquito population and have a long-lasting suppressive impact. An estimated 22% reduction in the fitness of YLE^{dsx} males, compared to wild-type males, was observed. Nonetheless, additional mathematical modelling that we have conducted to account for this fitness cost demonstrates that generational releases of YLE^{dsx}; *dsxF^{Δ11/+}* males would lead to the virtual collapse of the target population.

The results of this cage trial experiment are described in lines 261-297, Figure 5, and supplementary Figure 6. Space and capacity constraints prevented us from performing a parallel multigenerational experiment with an SIT system.

Minor points:

1) It is not exactly clear how the authors derived the dsxDF11 allele and whether it might have been isolated from the first set of experiments using the GFP-null dsx allele. Was the dsxDF11 allele obtained as an NHEJ induced copying error from those crosses?

We appreciate the valuable feedback from the reviewer. We described on lines 167-168 that this deletion was found while maintaining the YLE^{dsx} strains and we have added more detail to Figure 3a (and the figure caption) highlighting sequences around the cut site that might have triggered microhomology directed repair and led to the 11-base pair deletion.

2) The authors mention in the discussion that an important future issue to assess would be to estimate the rate at which functional NHEJ alleles might be generated by the YLE system. If they have any data pertaining to this important question it would be good to include it. For example, it would be good to know what fraction of NHEJ alleles they have sequenced are in-frame. It might also be good to discuss possible work arounds (in addition to those they already outline) in which the YLE element also expresses a second gRNA that targets a highly conserved dsx splice site (the same site that is targeted in the highly efficient gene-drives this same group has designed). Expressing this splice-site gRNA should generate a parallel recessive loss-of-function allele (no functional NHEJ alleles have observed to be generated at this site). This dual strategy might guarantee that all progeny were at least recessive sterile?

This is a valuable observation. We have expanded the discussion of this work to address in more detail the feasibility of multiplexing, either by using more than one gRNA or by targeting more than one gene at a time, making reference to emerging work in this field (lines 409-418).

Reviewer #2 (Remarks to the Author):

In this manuscript, Tolosana et al describe a Y-linked gene editing system capable of reducing the reproductive potential of *Anopheles gambiae* mosquitos. This follows closely from their previous work developing Cas9-based gene drive mosquitoes targeting the female-specific form of the doublesex gene. New components described in this paper are the source of Cas9, which has been moved out of the dsx locus (Kyrou et al 2018) and onto the Y-chromosome, and the development of a dominant negative deletion in dsx that induces intersex phenotypes. This contrasts with their previous disruption of dsx (insertion of EGFP), where heterozygous individuals formed normally, and only homozygous disruptions were intersex. Using these tools, the authors demonstrate that in the presence of paternally-supplied Cas9, conversion of WT dsx alleles to either EGFP or deletion alleles via gene drive is efficient. Modeling frameworks compare the efficiency of the approach to other self-limiting approaches, and suggest comparable or superior results. Overall, this is an important finding and would be of general interest, but several aspects of the manuscript are underdeveloped.

The authors utilize two Y-linked strains that exhibit different levels of gene drive, which the authors attribute to variable expression based on position. However, the authors do not attempt to measure expression levels or determine the position of the insertions.

We acknowledge the reviewers' comment on the speculation that the differences in the 'homing' of the Y-linked Cas9 in the two strains we analysed are due to variable expression based on position along the Y chromosome.

The Y chromosome of *Anopheles gambiae* is highly heterochromatic and repetitive, and to date, a full assembly for this chromosome is missing. The two Y-linked docking strains were independently generated using piggyBac transposon-mediated transposition, which inserts semi-randomly in the genome. Therefore, it is very unlikely that the transgene landed in the same locus. Since the YLE transgene inserted in the two docking strains is the same, a positional effect could account for the observed difference in Cas9 efficiency. Positional effects associated with differences in the expression/efficiency of transgenes have been widely described in *Anopheles* mosquitoes (Carballar-Lejarazu et al. 2013, Galizi et al. 2014, Galizi et al. 2016). The

complexity of the Y chromosome and the lack of assembly hinder sequencing experiments aimed at assessing the location of the two docking sites on the Y. Empirical studies such as fluorescence in situ hybridization (FISH) would need to be conducted to compare the insertions of the two docking sites. This work aimed at demonstrating the efficacy of the YLE and its potential for vector control. We acknowledge the value of these experiments, but they were beyond the scope of our current work. However, we do not dismiss the possibility of conducting future experiments, including FISH and experiments to measure Cas9 expression levels, in order to better define the characteristics of these strains.

The authors describe a second deletion (1bp) as also being potentially dominant negative, but make no attempt at discerning what aspects of these mutations lead to DN phenotypes. Do these maintain female-splicing but disrupt the ORF? Or alternatively, do these alone disrupt female splicing, and increase usage of the male form in XX individuals (leading to intersex)? As the strategy described by the authors depends on these phenotypes, as does the frequency by which new DN's arise, a basic understanding of this matter is highly relevant to the conclusions.

This is a valuable consideration. While it would be nice to know, the absence of this information does not distract from the key findings of the paper, which is a demonstration that two identified mutations in *dsx* are dominant and cause an intersex phenotype in females.

Both hypotheses mentioned by the reviewer are discussed in the manuscript as the potential mechanism leading to the dominant effect of these mutations, making reference to previous work on this gene (lines 393-399).

The authors limit their data collection to single female one generation outputs. It is somewhat surprising that they did not perform multi-generational studies as they have done in so many past studies given the model predictions that population-level effects should be rapid (10 generations for 80% reduction). If there are technical reasons as to why this is not possible/practical, what does this mean for the maturity of this approach?

Since reviewer 1 also raised a similar query, we set up cage trials to demonstrate and quantify this aspect. Please see response to reviewer 1.

Modelling assumes no fitness costs for YLE males, with only the fertility values demonstrated experimentally. While I understand it is not possible to perform every possible fitness test, I am surprised the modelling did not examine the tolerance of the approach for fitness costs in males (lifespan/mating competitiveness/developmental timing).

We appreciate the relevance of the reviewer's comment on this topic. During the multigenerational experiment we observed an estimated 22% reduction in the fitness of YLE^{dsx} males compared to wild-type males (Figure 5, Supp. Figure 7). However, our modelling shows that repeated releases of YLE^{dsx}; *dsxF^{Δ11/+}* males would lead to the virtual collapse of the target population.

Additionally, we included an extra panel in Figure 6 showing how the fitness cost associated to the YLE would affect the release rates required to suppress a target population.

Figure 2: Panel (a) shows a construct, but not how it is targeted to the Y chromosome. The details are in the methods, but it maybe more clear to indicate on the figure. Panel (c) refers to "modified dsx allele" and is colored in red, but really this is a transgenic EGFP allele. Showing this directly and labelling it as "dsxF-EGFPnull" would match the labels in panel (d) and make the figure much easier to follow. Panel (d) refers to colors (yellow, pink) that differ from the colors in the figure (maroon, red). Some symbols maybe did not convert correctly on the pdf (81.5 box 3.2 should be 81.5 +/- 3.2?) Dashed line is referred to in the legend but is not visible in the figure.

Thank you for this suggestion. Fig. 2a now shows a diagram showing how the constructs were integrated into the Y chromosome. The rest of the amendments highlighted here were also addressed.

Figure 3. Dominant negative females do not lay eggs, but do they feed? This is never stated, and contrasts with their previous paper (Kyrou et al 2018) where homozygous

dsx females were offered blood but did not feed. If they do feed, they could still be vectors even if sterile, so this is important to mention.

We appreciate that this is relevant information for using YLE as a tool for vector control. In response to the reviewer, we have now included an assay to test the ability of intersex individuals to feed on blood (only 27% of them are able to feed on blood) (lines 221-225, Fig. 3d).

Figure 4. dsxF11 allele and male YLE fertility. Symbols corrupted again. Colors referenced (yellow, pink) do not correspond to colors in figure.

This has been addressed.

Line 128: “dictated” is a strange term here. “facilitated”, “was advantageous” or something similar may be more accurate. Nobody forced you to choose dsx.

This paragraph has been modified, and this sentence is no longer shown as it was in the first version (lines 147-151).

Line 135-136: give reference for docking sites generated on Y chromosome. (lines 264-265: positions unknown?). I see the reference in the methods (23) for Line A , but Line B has no reference. Since that was the strain that performed the best, this is even more important to provide information on this site and the construct/method used to integrate it. Are the docking strains and YLE strains available for others to use? How can they be shared?

The references for both Y docking sites are now provided.

Line 141-143: why not measure Cas9 expression directly (mRNA or protein) to validate proposed expression differences?

While we acknowledge and understand that this would validate whether the differences on homing rates are due to differences in Cas9 expression, our aim was to develop a proof of concept for a YLE technology as a powerful self-limiting genetic control strategy and show its potential for vector control.

As mentioned above, we do not dismiss the possibility of conducting future experiments to measure Cas9 expression levels in order to better define the characteristics of these strains, but this aspect is out of the scope of the present study.